# Survey of potentially inappropriate prescriptions for common cold symptoms in Japan: A cross-sectional study

Yasuhisa Nakano[1], Takashi Watari[2,3]*, Kazuya Adachi[4], Kenji Watanabe[4], Kazuya Otsuki[1], Yu Amano[1], Yuji Takaki[4], Kazumichi Onigata[1,5]

**1** Faculty of Medicine, Shimane University, Shimane, Japan, **2** General Medicine Center, Shimane University, Shimane, Japan, **3** Division of Hospital Medicine, University of Michigan Health System, Ann Arbor, MI, United States of America, **4** Midori Pharmacy Co. Ltd., Shimane, Japan, **5** Postgraduate Clinical Training Center, Shimane University Hospital, Shimane, Japan

* wataritari@gmail.com

**Data Availability Statement:** The data that support the findings of this study are available for anyone from the General Medicine Center, Shimane University (contact information; tel +81-853-20-

## Abstract

### Background

Common cold is among the main reasons patients visit a medical facility. However, few studies have investigated whether prescriptions for common cold in Japan comply with domestic and international evidence.

### Objective

To determine whether prescriptions for common cold complied with domestic and international evidence.

### Methods

This cross-sectional study was conducted between October 22, 2020, and January 16, 2021. Patients with cold symptoms who visited the two dispensing pharmacies and met the eligibility criteria were interviewed.

### Main outcome measure

The pharmacists at each store and a physician classified the patients into two groups: the potentially inappropriate prescribing group and the appropriate prescribing group.

### Results

Of the 150 selected patients, 14 were excluded and 136 were included in the analysis. Males accounted for 44.9% of the total study population, and the median patient age was 34 years (interquartile range [IQR], 27–42). The prevalence rates of potentially inappropriate prescriptions and appropriate prescriptions were 89.0% and 11.0%, respectively and the median drug costs were 602.0 yen (IQR, 479.7–839.2) [$5.2 (IQR, 4.2–7.3)] and 406.7 yen (IQR, 194.5–537.2) [$3.5 (IQR, 1.7–4.7)], respectively. The most common potentially inappropriate prescriptions were the prescription of oral cephem antibacterial agents to patients

2217, e-mail: shimanegp@gmail.com) upon reasonable request.

**Funding:** YES T.W. is supported by grants from the National Academic Research Grant Funds (JSPS KAKENHI: 20H03913). The sponsor of the study had no role in the study design, data collection, analysis, or preparation of the manuscript.

**Competing interests:** The authors have declared that no competing interests exist.

who did not have symptoms of bacterial infections (50.4%) and β2 stimulants to those who did not have respiratory symptoms due to underlying disease or history (33.9%).

## Conclusions

Approximately 90% of prescriptions for common cold symptoms in the area were potentially inappropriate. Our findings could contribute to the monitoring of the use of medicines for the treatment of common cold symptoms.

## Introduction

Common cold symptoms are one of the main reasons patients visit a medical facility both internationally and in Japan [1, 2]. In the past, physicians mainly prescribed antimicrobial agents for common cold symptoms. Numerous studies have reported the actual practice of prescribing antimicrobial agents [3, 4] and the harmful effects of consumption of these antimicrobial agents by patients with common cold symptoms [5, 6]. According to the reports published in the United States, antimicrobials should not be prescribed for common cold as much as possible owing to drug resistance (more than 2 million antibiotic-resistant diseases are reported each year), the financial burden on the patient and healthcare system (approximately 50% of antimicrobial prescriptions are unnecessary, totaling to medical expenses of more than $3 billion annually), and side effects (5%–25% of patients who use antimicrobials experience adverse events) [7]. In Japan, approximately 60% of outpatients with common cold symptoms were prescribed antimicrobials in 2005 [3], which led to the establishment of the government-led Action Plan to address antimicrobial resistance in 2016.

Several drugs intended to treat common cold symptoms in adults have been useful in relieving symptoms. However, their efficacy is limited [8], and their benefits and disadvantages should be considered. Previous studies have indicated that non-steroidal anti-inflammatory drugs (NSAIDs) and first-generation antihistamines may be inappropriate for older patients and should not be prescribed frequently for common cold symptoms in this population [9]. However, there is a lack of evidence on whether physicians' prescriptions for common cold in Japan comply with the domestic and international evidence [10]. Therefore, we planned to conduct a study by examining prescriptions for common cold in Japan and conducting interviews with patients.

We aimed to investigate whether prescriptions for common cold at two dispensing pharmacies in Izumo City, Shimane Prefecture, Japan, comply with the domestic and international evidence and assess the appropriateness of the prescriptions from these pharmacies.

## Materials and methods

### Study design

This cross-sectional study was conducted for approximately 3 months, from October 22, 2020, to January 16, 2021. Two pharmacists from each of the two dispensing pharmacies in Izumo City conducted in-depth interviews with patients who had been prescribed medication to relieve cold symptoms.

## Study protocol

The study researchers were four experienced pharmacists and one physician who was an expert in primary care. The pharmacists collected the data directly, and the physician set the definition of common cold and the classification criteria for the two groups, namely, potentially inappropriate and appropriate prescribing groups, using data from previous studies based on domestic and international evidence (S1 Appendix) [11, 12]. Next, a questionnaire was prepared for collecting data from the patients, which included the following variables: patient sex and age, doctor's diagnosis and explanation of common cold, type of symptoms during the visit, duration of symptoms, presence of allergies, history, and current status of smoking, type of prescribed medication, and drug cost (S2 Appendix).

A flowchart of the patient selection process is shown in Fig 1. The patients visited a hospital or clinic, received a prescription, and then visited the pharmacy to pick up the medication. Next, the pharmacist assessed whether the patient met the inclusion criteria using the questionnaire (S2 Appendix) and confirmed the patients' symptoms and details of the prescribed medication (the content of the questionnaire was based on the patients' subjective opinions) (n = 150). The physician, who was not the prescribing physician, excluded patients who met the exclusion criteria, after which two pharmacists and a physician classified the patients with prescriptions into two groups: the potentially inappropriate prescribing group and the appropriate prescribing group (S1 Appendix) (N = 136).

## Definitions

The term "potentially inappropriate prescribing" in this study refers to prescribing of medicines for which there is no clear evidence or indication or that are not suitable for the patient's symptoms or are contraindicated or administered in inappropriate doses, all of which indicate that the disadvantages of prescribing are likely to outweigh the benefits [11, 12]. "Appropriate prescription" refers to prescribing drugs suitable for the patient's symptoms where the benefits of prescribing for the patient's condition are likely to outweigh the disadvantages for various reasons and the prescription is based on clear evidence and indications. The two pharmacists and the primary care expert physician made these judgments using the classification criteria developed in previous studies (S1 Appendix). The term "common cold symptoms" refers to one or more common cold symptoms, namely, acute low-grade fever, headache, cough symptoms, nasal congestion symptoms, and sore throat symptoms [1, 13]. ※ S1 Appendix includes reference [14–33].

## Inclusion criteria

In this study, we included adult patients aged 20–80 years who reported that they had one or more typical symptoms of common cold, namely, low-grade fever, headache, cough, nasal congestion, and sore throat symptoms [1, 13] and that these symptoms were comparable to those they had previously. Prescriptions were included only for drugs intended to relieve cold symptoms (drugs included in S1 Appendix). We also ensured that the medications prescribed to the patients were their own medications.

## Exclusion criteria

Patients with underlying diseases and those who were pregnant or breastfeeding were excluded. Since there is not enough scientific evidence for Chinese herbal medicines, also called "Kampo medicines," prescriptions containing Kampo medicines were excluded from this study. Kampo medicine is a combination of herbal medicines made from a number of

plants and minerals originally developed in China. It was introduced in Japan and developed into a unique treatment method in Japan. Through thousands of years of experience, the effects of various combinations of herbal medicines have been confirmed.

Patients who regularly took medicines other than those to relieve cold symptoms were also excluded to reduce the impact or interaction of other medicines on the results of our study. Patients with incomplete data, such as incomplete interviews, and patients who were judged by the prescribing physician or the physician involved in the study as not having evident symptoms of common cold were excluded from the study.

## Outcome measures

The primary outcome was the frequency of "potentially inappropriate prescription " and "appropriate prescription." Other outcomes included the cost of each prescription drug (Japanese yen was converted to US dollars based on the exchange rate of 114.92 yen per 1 US dollar on December 29, 2021) in "potentially inappropriate prescriptions" and "appropriate prescriptions," types of symptomatic and antimicrobial drugs prescribed, and types of prescription drugs used for classifying prescriptions into the potentially inappropriate prescribing group based on patient interviews and prescription drug evidence.

## Statistical analyses

We used standard descriptive statistics to calculate the number, percentage, median, and IQR (interquartile range). All statistical analyses were performed using JMP®, Version 15. SAS Institute Inc., Cary, NC, 1989–2021.

## Ethics approval

The present study was conducted after obtaining approval from the Ethics Committee of the Shimane University School of Medicine (No. 20200622–2. Approval date: October 19, 2020). The included patients provided informed consent to participate in this study.

# Results

## Sample characteristics

The study included 150 patients with symptoms such as fever, headache, cough, nasal congestion, sore throat without underlying disease. Among them, 14 were excluded because the physician, who was not the prescribing physician, judged that the patients did not have cold symptoms, or the interview was incomplete (Fig 1). Patient characteristics are presented in Table 1. The median patient age was 34 years (interquartile range [IQR], 27–42), and the median duration of common cold symptoms was 2 days (IQR, 1.0–4.8). For 89.7% of the patients, the physician described the symptoms as "cold, upper respiratory tract infection, pharyngitis, and swollen tonsils." Of all the patients, 18.4% had some kind of allergy, and 11.0% were smokers with median pack-years of 7.5% (IQR, 5–16).

## Patients' symptoms

Table 2 shows the details of the patients' symptoms. The most common symptom reported by the patients was sore throat (63.2%), followed by cough (52.2%), runny nose (46.3%), phlegm (41.9%), nasal congestion (36.8%), lethargy (35.3%), headache (30.1%), fever (28.7%), painful swallowing (26.5%), sneezing (18.4%), and abdominal pain (1.5%). The median body temperature of the patients who had a fever was 37.4˚C (IQR, 37.1–38.0).

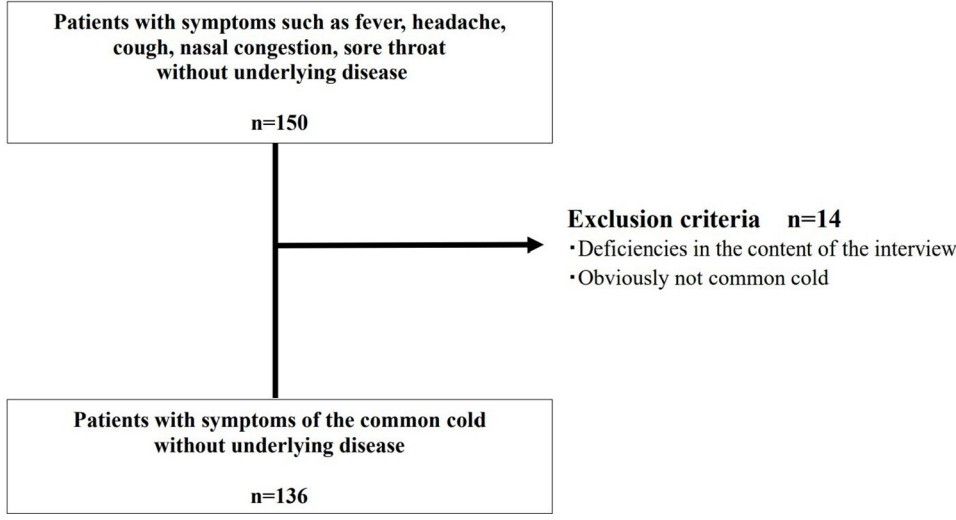

**Fig 1. Flowchart of the patient selection process.**

## Details of prescriptions

Among all the prescriptions (n = 136), the most commonly prescribed symptomatic medications were H1 receptor antagonists (chlorpheniramine maleate, bepotastine; 41.9%), followed by expectorants (carbocysteine, bromhexine, ambroxol; 41.2%), NSAIDs (diclofenac, thiaramide, loxoprofen; 40.0%), β2-stimulants (mainly tulobuterol patches, terbutaline; 36.0%) and decalinium chloride lozenges (indicated for sore throat in Japan; 36.0%), non-narcotic antitussives (dimethorphan, dextromethorphan, tipepidine; 22.1%), tranexamic acid (indicated for sore throat in Japan; 19.9%), acetaminophen (18.4%), multi-ingredient cold medication (non-pyrine cold remedy combination granules, pyrazolone antipyretic analgesic anti-inflammatory combination granules; 7.4%), and leukotriene receptor antagonists (pranlukast, montelukast; 3.7%). Regarding antimicrobial agents, 44.9% of all prescriptions contained oral cephem (cefcapene pivoxil, cefdinir), 25.0% contained new quinolones (galenoxacin, levofloxacin), 9.5% contained macrolides (azithromycin, clarithromycin), and 2.2% contained penicillin (amoxicillin) (Table 3).

## Details of potentially inappropriate prescription

The breakdown of potentially inappropriate prescription drugs in terms of each prescribed drug is shown in Table 3. β2 stimulants (83.7%), followed by NSAIDs (11.3%), had the highest percentage among symptomatic medicines. Among the antimicrobial agents, oral cephems and new quinolones had the highest rates (100%), followed by macrolides (92.3%).

**Table 1. Background characteristics of patients (n = 136).**

| Male sex, No (%) | | 61 (44.9) |
|---|---|---|
| Age, median (IQR), y | | 34 (27–42) |
| Days of cold symptoms (IQR), d | | 2.0 (IQR 1.0–4.8) |
| Patients who received any explanation from a doctor, No (%) (Ex. common cold, upper respiratory tract inflammation, sore pharyngitis, swollen tonsils) | | 122 (89.7) |
| Any allergies, No (%) | | 25 (18.4) |
| Smoking | n (%) Pack-years, median (IQR) | 15 (11.0) 7.5 (5–16) |

IQR, interquartile range.

**Table 2. Patients' symptoms.**

|  | n | % |
|---|---|---|
| Sore throat | 86 | 63.2 |
| Cough | 71 | 52.2 |
| Runny nose | 63 | 46.3 |
| Phlegm | 57 | 41.9 |
| Nasal congestion | 50 | 36.8 |
| Lethargy | 48 | 35.3 |
| Headache | 41 | 30.1 |
| Fever | 39 | 28.7 |
|  | 37.4°C | (IQR 37.1–38.0) |
| Painful swallowing | 36 | 26.5 |
| Sneezing | 25 | 18.4 |
| Abdominal pain | 2 | 1.5 |
| Others | 17 | 12.5 |

IQR, interquartile range.

The percentage of each potentially inappropriate drug in the total number of potentially inappropriate prescriptions is shown in Table 3. β2 stimulants (33.9%), followed by NSAIDs (5.0%), had the highest percentage among symptomatic medicines. Among antimicrobial agents, oral cephem had the highest percentage (50.4%), followed by new quinolones (28.1%) and macrolides (10.0%).

There were two contraindicated doses (for patients with a history of β2-stimulant hypersensitivity), and the pharmacist posed questions regarding the prescription. There were no dosage errors.

## Rates of potentially inappropriate prescription and medication costs

When the prescriptions were classified using the classification criteria (S1 Appendix), 11.0% (n = 15) fell under the appropriate prescription group and 89.0% (n = 121) under the potentially inappropriate prescription group (Fig 2). The median total cost of the prescribed drugs was 593.6 yen (IQR, 470–795.6) [$5.2 (IQR, 4.1–6.9)]. In particular, the median cost of drugs in the potentially inappropriate prescription group was 602.0 yen (IQR, 479.7–839.2) [$5.2 (IQR, 4.2–7.3)], and the median cost of drugs in the appropriate prescription group was 406.7 yen (IQR, 194.5–537.2) [$3.5 (IQR, 1.7–4.7)].

## Discussion

This study suggests that approximately 90% of the prescriptions for common cold symptoms in this area are inappropriate and that antimicrobials and symptomatic drugs, such as β2-stimulants, which are not suitable for the respiratory symptoms in the absence of underlying disease or history, maybe inappropriately prescribed. This is the first cross-sectional study in Japan to investigate whether prescriptions for common cold comply with domestic and international evidence.

### The rationale for classification as the potentially inappropriate prescribing group

Oral cephem antibiotic, β2-stimulants, and new quinolone antibacterials were the top three drugs used in classifying prescriptions into the potentially inappropriate prescribing group, based on patient interviews and prescribed drugs and prescription drug evidence. The

**Table 3. Details of prescription drugs (n = 136).**

| | Breakdown of all prescriptions | | Breakdown of potentially inappropriate prescription drugs for each drug prescribed. | | Percentage of potentially inappropriate prescriptions drugs for each drug in the total potentially inappropriate prescriptions (n = 121). |
|---|---|---|---|---|---|
| | n | % | n | % | % |
| **Symptomatic medicine** | | | | | |
| H1 receptor antagonist | 57 | 41.9 | 0 | 0 | 0 |
| Expectorant drug | 56 | 41.2 | 0 | 0 | 0 |
| NSAIDs | 53 | 40.0 | 6 | 11.3 | 5.0 |
| β2-stimulant | 49 | 36.0 | 41 | 83.7 | 33.9 |
| Decalinium chloride (cough drop) | 49 | 36.0 | 0 | 0 | 0 |
| Non-narcotic antitussive | 30 | 22.1 | 0 | 0 | 0 |
| Tranexamic acid | 27 | 19.9 | 0 | 0 | 0 |
| Acetaminophen | 25 | 18.4 | 0 | 0 | 0 |
| Multi-ingredient cold medication | 10 | 7.4 | 0 | 0 | 0 |
| #Non-pyrine cold remedy combination granules, Pyrazolone antipyretic analgesic anti-inflammatory combination granules | | | | | |
| Leukotriene receptor antagonist | 5 | 3.7 | 0 | 0 | 0 |
| Others | 13 | 9.4 | 3 | 23.1 | 2.5 |
| **Antibacterial drugs** | | | | | |
| Cephem (Oral) | 61 | 44.9 | 61 | 100 | 50.4 |
| New quinolone | 34 | 25.0 | 34 | 100 | 28.1 |
| Macrolide | 13 | 9.5 | 12 | 92.3 | 10.0 |
| Penicillin | 3 | 2.2 | 0 | 0 | 0 |
| Cephem (Nasal spray) | 1 | 0.7 | 0 | 0 | 0 |

NSAID, non-steroidal anti-inflammatory drug.

Note: Active ingredients in 1 g of Non-pyrine cold remedy combination granules (salicylamide, 270 mg; acetaminophen, 150 mg; caffeine anhydrous, 60 mg; promethazine methylene disalicylate, 13.5 mg); active ingredients in 1 g of Pyrazolone antipyretic analgesic anti-inflammatory combination granules (isopropylantipyrine, 150 mg; acetaminophen, 250 mg; allyl isopropylacetylurea, 60 mg; caffeine anhydrous, 50 mg).

rationale for classifying prescriptions of oral cephem antibiotics as potentially inappropriate is as follows: antimicrobial agents are not beneficial for common cold symptoms, and their side effects are notable [15]. In the case of suspected streptococcal infection, penicillin is recommended as the first choice because of its proven efficacy and safety, narrow spectrum, and low cost [31]. The rationale for classifying prescriptions containing β2-stimulants as potentially

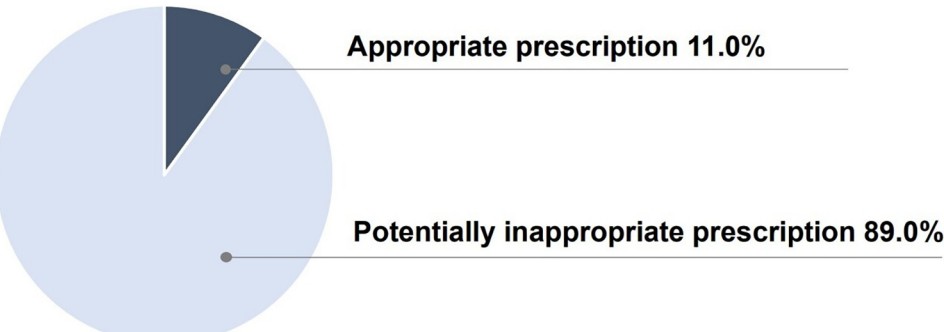

**Fig 2. Classification of prescriptions based on criteria.**

inappropriate is as follows: β2-stimulants are useful in patients with a history of cough, asthma, chronic obstructive pulmonary disease (COPD), and other diseases [27]. However, there is no clear benefit of these drugs for acute bronchitis and acute cough in adults, and side effects such as tremor and neurological symptoms may occur [28]. The tulobuterol patch, which was mainly used in this study, is approved for use only in Japan and Korea, and there is insufficient evidence in terms of its efficacy and safety [29]. Considering this information, we concluded that it is reasonable to consider a prescription for β2-stimulants for patients without underlying diseases such as asthma or COPD and with strongly suspected findings for acute cold symptoms as potentially inappropriate. The rationale for judging new quinolones as inappropriate is as follows. New quinolones are considered inappropriate as they are ineffective against viruses which cause common cold. The risk of aortic aneurysm and aortic dissection by inducing collagen degradation has been reported for some new quinolones [32]. Given the magnitude of the risk of adverse effects, it is unlikely that they need to be administered to patients with acute cold symptoms.

## Factors responsible for prescriptions of potentially inappropriate drugs

Based on previous studies, four major factors may be involved in the prescriptions becoming potentially inappropriate: The first factor is that patients may be seeking medications from doctors [34]. The Japanese healthcare system provides easy access to higher medical institutions and clinics. Due to the low cost of medical care in Japan, patients may be inclined to receive some kind of medication and feel relieved [35]. The second factor may be the physician's attitude toward consultation. For example, doctors intend to satisfy patients by prescribing them medications [36] and quickly end the consultation [37]; their prescribing patterns are fixed to some extent according to the age and sex of the patient [38]. The third factor is the lack of evidence on physicians' standard of care for the treatment of common cold. In Japan, despite the publication of national guidelines in 2016 regarding the appropriate use of antimicrobials, there was no significant change in the trend of antimicrobial use among outpatients with acute upper respiratory tract infection before and after the publication [39]. In this study, the percentage of antimicrobials in prescriptions (83%) was higher than that reported in previous studies (60%) [3]. Furthermore, third-generation oral cephem antibacterial agents were used in approximately 50% of the prescriptions for common cold symptoms, followed by new quinolone and macrolide antibacterials, and the proportion of prescriptions was approximately the same as that in previous studies [3]. This information suggests that there is insufficient awareness regarding the need to refrain from prescribing unnecessary antimicrobials for common cold symptoms. Fourth, there is a lack of primary care education for physicians. It is common for Japanese primary care physicians to specialize in organ-specific treatment and start private practices, taking on the role of primary care providers in their respective regions [40]. In Japan, many physicians start their own clinical practice without undergoing any special training for minor illnesses [41]. Therefore, it is possible that many physicians in this region also routinely prescribe the same drugs that they prescribe for organ-specific treatment. However, it is unlikely that any single solution will alter the prescribing habits of physicians [42]. Cultural factors may also be involved in addressing this problem, and steps other than continuing medical and patient education are necessary.

## Pharmacists' views on potentially inappropriate prescription

In Japan, the importance of the pharmacists' intervention in prescribing has been highlighted in recent years [43]. Pharmacists in Japan are proactive in making inquiries regarding the "safety and dosage volume" of drugs [43]. In contrast, in cases where there is a possibility of

inappropriate prescribing but the pharmacist is unsure about making an inquiry, the pharmacist often does not make an inquiry, taking the prescribing physician's intentions into consideration. One of the reasons underlying this is that a certain number of physicians consider the pharmacist's frequent inquiries to be bothersome, and pharmacists, who place importance on a good relationship with the local physician, are hesitant to make such inquiries. Such psychological barriers between pharmacists and doctors may be detrimental to patients. This is an issue that needs to be gradually addressed in the future.

## Considerations for drug costs

The median cost of drugs for potentially inappropriate prescriptions and appropriate prescriptions were 602.0 yen (IQR, 479.7–839.2) [$5.2 (IQR, 4.2–7.3)] and 406.7 yen (IQR, 194.5–537.2) [$3.5 (IQR, 1.7–4.7)], respectively. As shown in the previous study, "Japan's universal health insurance system covers all prescription drugs generally; therefore, patient co-payments are minimal, and this is a factor that contributes towards doctors prescribing drugs easily" [44, 45]. This makes it easier for doctors to prescribe unnecessary drugs leading to a higher cost of drugs. As part of the Choosing Wisely Campaign, it is generally recommended that patients ask their doctors the following five questions [46]. (1) Do I really need this test or procedure? (2) What are the risks? (3) Are there simpler, safer options? (4) What happens if I do not do anything? (5) What are the costs? Patients must be aware of their own "treatment" by referring to the above questions to promote the appropriate use of medicines in Japan. It is necessary to incorporate education and measures that consider the interactive opinions of the medical professionals and patients.

## Limitations

This study had several limitations. First, the sample size was small, and the survey was limited to hospitals and clinics in Izumo City; therefore, there may have been strong influences of the characteristics of doctors in the area, such as biases in prescribing. Second, because the survey was conducted in a dispensing pharmacy, the actual conversations between the doctors and patients in the clinic, patients' physical findings, and information in their medical records were not captured. Third, the study excluded prescriptions of Kampo medicines. Since Kampo medicines are often prescribed for common cold symptoms in Japan, the results of this study do not fully reflect the actual situation of prescriptions for common cold symptoms in Japan. Fourth, the criteria used to classify prescriptions were not complete. Several previous overseas studies have evaluated the efficacy of cold medicines; however, the efficacy of many of these medicines is unclear, which may have led to the misclassification of the prescriptions. Fifth, there may have been influences of changes in the epidemiology of common cold caused by the novel coronaviruses. Fortunately, however, during the study period (October 2020 to January 2021), we had the lowest incidence of coronavirus disease in Japan, with the lowest positivity rate in polymerase chain reaction (S3 Appendix). Therefore, although we believe that the impact of the epidemiological changes caused by the new coronavirus infection in this study was quite limited, we cannot deny the possibility of its impact on the study. Sixth, the details regarding the prescribing physician were not investigated in this study as the Ethics Committee stipulates that "the content that identifies the prescribing physician (specialty, affiliation, and the number of individuals) must not be added to the content of the interview because it may cause disadvantages to the prescribing physician." We believe that the prescribing physician's details are important factors and further research is needed. Lastly, this study was limited to adult patients with no underlying medical conditions. The content of prescriptions for common cold patients is likely to be affected by the presence or absence of underlying diseases

and the age of the patients. Therefore, the results of this study may not fully reflect the current status of prescribing for common cold symptoms in Japan.

## Conclusion

Approximately 90% of the prescriptions for common cold symptoms in this local area were potentially inappropriate, and antimicrobials and symptomatic drugs such as β2-stimulants, which are not suitable for the patients' respiratory symptoms in the absence of underlying diseases or history, may be inappropriately prescribed. This finding is expected to help promote the appropriate use of medicines for common cold symptoms and improve the quality of medical care. However, this was a pilot study at the regional level, and it is necessary to conduct a nationwide survey in the future.

## Supporting information

**S1 Appendix. Classification criteria.**
(XLSX)

**S2 Appendix. Patient information sheet.**
(DOCX)

**S3 Appendix. Number of people newly infected with SARS-CoV-2 in Izumo city.**
(PPTX)

## Acknowledgments

We are very grateful to Midori Pharmacy Co. Ltd. for their cooperation in interviewing the target patients.

## Author Contributions

**Conceptualization:** Yasuhisa Nakano, Takashi Watari.

**Data curation:** Yasuhisa Nakano, Takashi Watari, Kazuya Adachi, Kenji Watanabe, Kazuya Otsuki, Yu Amano.

**Formal analysis:** Yasuhisa Nakano, Takashi Watari, Kazuya Otsuki, Yu Amano, Yuji Takaki.

**Funding acquisition:** Takashi Watari.

**Investigation:** Yasuhisa Nakano, Kazuya Adachi, Kenji Watanabe, Yuji Takaki.

**Methodology:** Yasuhisa Nakano, Takashi Watari, Kenji Watanabe, Kazuya Otsuki, Yuji Takaki.

**Project administration:** Yasuhisa Nakano, Takashi Watari, Kazuya Adachi.

**Resources:** Takashi Watari, Kazuya Adachi, Kenji Watanabe, Kazumichi Onigata.

**Software:** Kazuya Otsuki.

**Supervision:** Takashi Watari, Kazuya Otsuki, Yu Amano, Kazumichi Onigata.

**Validation:** Takashi Watari, Kazuya Adachi.

**Writing – original draft:** Yasuhisa Nakano, Takashi Watari.

**Writing – review & editing:** Takashi Watari.

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
