## [Decision Letter · Decision Letter 0]

6 Dec 2021

PONE-D-21-33168Survey potentially inappropriate prescriptions for common cold symptoms in Japan: A prospective observational studyPLOS ONE

Dear Dr. Watari,

Thank you for submitting your manuscript to PLOS ONE. After careful consideration, we feel that it has merit but does not fully meet PLOS ONE’s publication criteria as it currently stands. Therefore, we invite you to submit a revised version of the manuscript that addresses the points raised during the review process.

The manuscript does not reach to an enough level for the acceptance in PlosOne in the present form.

See the Reviewers' comments carefully and respond them appropriately.

We look forward to receiving your revised manuscript.

Kind regards,

Masaki Mogi

Academic Editor

PLOS ONE

Journal Requirements:

6. Please include a separate caption for each figure in your manuscript.

Reviewers' comments:

Reviewer's Responses to Questions

**Comments to the Author**

1. Is the manuscript technically sound, and do the data support the conclusions?

Reviewer #1: Yes

Reviewer #2: Yes

2. Has the statistical analysis been performed appropriately and rigorously? 

Reviewer #1: Yes

Reviewer #2: No

3. Have the authors made all data underlying the findings in their manuscript fully available?

Reviewer #1: Yes

Reviewer #2: No

4. Is the manuscript presented in an intelligible fashion and written in standard English?

Reviewer #1: Yes

Reviewer #2: Yes

5. Review Comments to the Author

Reviewer #1: I find the manuscript to be well written, informative and concise. The paper presents novel findings based on analysis of patient interviews and prescriptions, which will be of interest to the readers.

Reviewer #2: The authors report an evaluation of the appropriateness of prescriptions for the common cold. The authors interviewed 136 patients presented to a pharmacy in Japan and evaluated the appropriateness and costs associated with the prescriptions provided.

Comments

1. Line 37 – remove word “Even”

2. Line 42 – Remove “on the contrary” as the statement implies similarity (60% vs 50%) and not “contrary”

3. Line 56 – suggest rephrase objectives statement. What is meant by “to determine the actual status of prescriptions” is unclear. Suggest state that the objective was to assess the appropriateness of prescriptions

4. S1 Appendix: The appendix provides a summary of the evidence used for the authors to adjudicate the prescriptions. However, the way the authors used this evidence summary to adjudicate the prescriptions is a bit opaque. It is not clear to me how each prescription was classified. Can the authors add another column clarifying which medications were appropriate or inappropriate based on the criteria used? For example; amoxicillin described as first line for GAS pharyngitis. How did the authors adjudicate amoxicillin or penicillin prescriptions for patients with pharyngitis without swab confirmation of GAS? I did not see a section for swab results on the interview form.

5. Please clarify what Cephem is? I am not familiar with this from North America. In the appendix listed as a nasal spray but in the manuscript it is treated as a systemic antibiotic. If a topical spray I suggest separating out from the systemic antimicrobials

6. Can the authors clarify what proportion of “inappropriate prescriptions” were for incorrect dose and which were unnecessary?

7. Line 126: I am not familiar with “Kampo medicine”, can the authors briefly explain this exclusion?

8. Line 142: I am not sure why the primary outcome was only presented descriptively and the secondary outcome was analyzed with a statistical test. With 136 participants it is a missed opportunity to not evaluate for predictors of inappropriate prescribing from the data collected. I recommend the authors perform bivariate and multivariable analysis evaluating predictors for inappropriate prescriptions.

9. I was surprised the authors did not capture whether a throat swab (rapid antigen or culture) was done for GAS. Is this common practice in primary care in Japan? Given how common sore throat as a complaint was I am not clear how antibiotic prescriptions were adjudicated without this information. Were all antibiotics assumed to be inappropriate? This relates back to point 4.

10. Line 200: Are these percentages of antibacterial agents the percent of all prescriptions? Percent of antimicrobials? Or percent of patients?

11. As above comment if Cephem is not a systemic antibiotic consider separating this out from the other antibiotics

12. Table 3 lists the percent of prescriptions. Can you add the percent inappropriate to this table? For example 41.9% of prescriptions were H1 blockers (or is it 41.9% of patients received H1 blocker?)…what percent of those 57 H1 blockers were inappropriate?

13. Can you convert Yen to US dollars in brackets?

14. Table 4: Suggest combine with table 3 and include numbers (not just percentages). How many prescriptions, what percent of total, and what number and percent were inappropriate

15. As per point 8 suggest include a statistical evaluation of predictors of inappropriateness

16. Line 260: clarify if Cephem is topical or systemic

17. Line 261: “Penicillin is recommended as the first choice…” how was this factored into the study? It is not clear how penicillin was adjudicated in this study (appropriate or inappropriate?)

18. Line 273: The provided reason for classifying quinolones as inappropriate is not accurate. Quinolones are inappropriate because they have no effect on viruses which cause the common cold. You have listed 1 rare but important side effect of that medication class.

19. Line 280: I don’t understand the separation of sections titled “Factors leading…” and “Factors responsible…” The authors discuss one potential aspect leading to inappropriate prescribing (medical education). I agree this may be a factor but there is a whole behavioral science literature of the many complex reasons for inappropriate prescribing – perceived patient expectations, fear, habit, etc. Many reasons beyond education and knowledge base. While this may be beyond the scope of this manuscript to discuss in detail these are important to briefly discuss as the solutions to this complex problem will involve much more than modifying medical education or continuing medical education as we know these interventions, while important, have overall limited impact on behavior change and quality improvement.

20. Line 291: The authors summarize literature on drivers of inappropriate drugs but missed an opportunity in this study to contribute to that literature. As per point 8 and 15 above I suggest you add that evaluation.

21. Line 321: Suggest add to limitations that the findings are limited to adults without underlying medical comorbidities (as per study inclusion criteria)

6. PLOS authors have the option to publish the peer review history of their article (what does this mean?). If published, this will include your full peer review and any attached files.

Reviewer #1: No

Reviewer #2: No

---

## [Author Response · Author response to Decision Letter 0]

21 Jan 2022

January 13, 2022

Dear PLOSONE Editor:

Thank you for giving us the opportunity to revise the manuscript. The constructive suggestions and feedback provided by the Reviewer have enabled us to substantially improve the quality of our paper. We have provided a point-by-point response to the Reviewer’s suggestions. The changes have been indicated in a yellow highlight in the revised manuscript.

Regarding open data, we have indicated that the data from this study are available upon request. This is because our study was approved by the Ethics Committee of the Shimane University School of Medicine (No. 20200622-2. Approval date: October 19, 2020), as a pilot study to investigate the inappropriateness of prescribing by doctors in Japan. Before launching the study, the open publication of this information was moderately restricted by the IRB because these are potentially sensitive information for the prescribing doctors, and public disclosure of this data could have a strong impact on the management of the private clinics and practice. We believe, therefore, that this study is important and should be used as a foundation to expand future research on inappropriate prescribing that may be prevalent throughout Japan. 

Once again, thank you for your thorough and supportive peer review.

On behalf of all the authors, yours sincerely,

Corresponding author

Takashi Watari, MD, MHQS, MCTM, Ph.D 

Shimane University Hospital, General Medicine Center, Shimane, Japan

89-1, Enya-cho, Izumo shi, Shimane, 693-8501, Japan

e-mail: wataritari@gmail.com; Phone: +81-853-20-2005; Fax: +81-853-20-2375

Reviewer 

1. Method

I would suggest adding more in the Statistical analyses section on the statistical methods used for deriving results described in the results section (including ones included in the Tables). For example, currently there is no mention on IQRs in the statistical analyses section – please add an explanation on this. 

Response: 

We appreciate and agree with the reviewer’s comment. The following changes have been made in the revised manuscript:

Changes (Method_Statistical Analyses)

P7, L148 

Background: We used standard descriptive statistics to calculate the number, percentage, median, and IQR (interquartile range). All statistical analyses were performed using JMP®, Version 15. SAS Institute Inc., Cary, NC, 1989–2021.

Reviewer

2. Results

How many physicians have prescribed the drugs in this time? What are the specialties of the doctors who prescribed it? What percentage of prescribing physicians belong to a hospital or clinic? Please add their characteristics if possible.

Response

Thank you for your valuable questions. As mentioned, we postulated that the specialty and affiliation of the prescribing physician is a very important factor in considering the results of this study. However, the Ethical Review Committee of the Shimane University School of Medicine (Approval No. 20200622-2, Approval date: October 19, 2020) has issued an order to the effect that "the content that identifies the prescribing physician (the prescribing physician's specialty, affiliation, and number of prescribers) must not be added to the content of the interview as it may be disadvantageous for the prescribing physician. Therefore, we cannot include the prescribing physicians’ specialty, affiliation, and number of prescribers since we have not interviewed them for these points. This information has been added to the Limitations section.

Addition (Discussion_ Limitation)

P18, L347 to P19, L359 (marked_manuscript)

Limitaion: Sixth, the details regarding the prescribing physician were not investigated in this study as the Ethics Committee of the Shimane University School of Medicine (No. 20200622-2. Approval date: October 19, 2020) stipulates that "the content that identifies the prescribing physician (specialty, affiliation, and the number of individuals) must not be added to the content of the interview because it may cause disadvantages to the prescribing physician.” We believe that the prescribing physician's specialty, affiliation, and the number of individuals are important factors while considering the results of this study. Therefore, further research is needed. 

Reviewer

3.Discussion

1)Four issues on the part of physicians have been pointed out as factors in inappropriate prescribing (Cephem antibacterials, β2-stimulants, and new quinolone antibacterials). What about the role of pharmacists, such as their intervention against inappropriate prescriptions? For example, in Japan, as a countermeasure against polypharmacy, a pharmaceutical management fee has been newly established as a dispensing fee, and pharmacists are encouraged to propose drug reduction.

Since there is no pharmacist's view for inappropriate prescribing in this document, please add their perspective.

Response

Thank you for your valuable comment. As pointed out, a statement regarding the pharmacist's perspective was not included in the manuscript. Therefore, the following statement has been added to the revised manuscript:

Addition (Discussion_ Pharmacists' views on potentially inappropriate prescription)

P17, L306 

In Japan, the importance of the pharmacists' intervention in prescribing has been highlighted in recent years [43]. Pharmacists in Japan are proactive in making inquiries regarding the "safety and dosage volume" [43]. In contrast, in cases where there is a possibility of inappropriate prescribing but the pharmacist is unsure about making an inquiry, the pharmacist often does not make an inquiry, taking the prescribing physician's intentions into consideration. One of the reasons underlying this is that a certain number of physicians consider the pharmacist's frequent inquiries to be bothersome, and pharmacists, who place importance on a good relationship with the local physician, are hesitant to make such inquiries. Such psychological barriers between pharmacists and doctors may be detrimental to patients. This is an issue that needs to be gradually addressed in the future.

Reviewer

3.Discussion

2)It has been shown that medication costs are higher in the inappropriate prescription 

group, but this point was not good enough explained, though there are some words about Japanese healthcare system in the “Discussion”

Response

Thank you for bringing this matter to our attention. We have added the following information as suggested:

Changes (Discussion_ Considerations for drug costs)

P17, L322 (marked_manuscript)

As shown in the previous study, "Japan's universal health insurance system covers all prescription drugs without restrictions; therefore, patient co-payments are minimal, and this is a factor that causes doctors to prescribe drugs easily”［44,45］, making it easier for doctors to prescribe unnecessary drugs leading to a higher cost of drugs.

Reviewer 

3.Discussion

3)It was described the first factor is that patients may be seeking medications from doctors in this document. However, some authors interviewed the patients actually. Did they confirm this factor? What was the patients' view?

 From the perspective of “Choosing Wisely”, how do the authors think about the health literacy education for patients?

Response 

Thank you for your insightful comment. In this study, we did not check whether the patients requested their doctors to prescribe medications as we did not include it in the survey items. Therefore, we do not know what the patients' exact intentions were behind seeking prescriptions.

In Australia, the following CWs are being promoted among the patients and the public in a campaign called "Ask Your Doctor Five Questions"(Michiko Yamamoto, Activities of Choosing Wisely and Role of Pharmacists, Yakugaku Zasshi. 2019;139(4):551-556.). (1) Is this test or procedure necessary for me? (2) What risks or side effects does it have? (3) Is there a simpler and safer option? (4) What will happen if I do not take the treatment? (5) How much does it cost? (5) How much will it cost and will my health insurance cover it? 

It is necessary for patients to be aware of their own "treatment" by referring to the above questions to promote the appropriate use of medicines in Japan. It is necessary to incorporate education and measures that take into account the interactive opinions of the medical professionals and patients.

――――――――――――――――――――――――――――――

Reviewer

1. Line 37 – remove word “Even”

Response

Thank you for bringing this error to our attention. We have made the following change accordingly.

Changes (Introduction)

P2(marked_manuscript):”Even” has been removed.

Reviewer

2. Line 42 – Remove “on the contrary” as the statement implies similarity (60% vs 50%) and not “contrary”

Response

Thank you for bringing this error to our attention. We have made the following change accordingly.

Changes (Introduction)

P2 (marked_manuscript): ” On the contrary” has been removed.

Reviewer

3. Line 56 – suggest rephrase objectives statement. What is meant by “to determine the actual status of prescriptions” is unclear. Suggest state that the objective was to assess the appropriateness of prescriptions

Response

Thank you for your valuable suggestion. We have rephrased the sentence accordingly.

Changes (Introduction)

P3, L59 (marked_manuscript): assess the appropriateness of prescriptions

Reviewer

4. S1 Appendix: The appendix provides a summary of the evidence used for the authors to adjudicate the prescriptions. However, the way the authors used this evidence summary to adjudicate the prescriptions is a bit opaque. It is not clear to me how each prescription was classified. Can the authors add another column clarifying which medications were appropriate or inappropriate based on the criteria used? For example; amoxicillin described as first line for GAS pharyngitis. How did the authors adjudicate amoxicillin or penicillin prescriptions for patients with pharyngitis without swab confirmation of GAS? I did not see a section for swab results on the interview form.

Response

Thank you for bringing this matter to our attention. As you indicated, we have added the rationale for the classification of the prescription in the column G of “appendix1___classification_criteria” 

The decision to prescribe amoxicillin is described below. In primary care settings in Japan, the GAS pharyngeal swab is often omitted; the patient's risk of streptococcal infection is assessed based on the Center criteria, and if the risk is high, amoxicillin is generally prescribed without the GAS pharyngeal swab. In this case, when the pharmacist interviewed the patient, it was confirmed that the patient had been told by the physician that there was a high possibility of pharyngitis or streptococcal infection (appendix 2 item (13)) and if the patient has symptoms such as fever and sore throat and amoxicillin is prescribed, it is considered an appropriate prescription. This information has been described in the column G (penicillin) of “appendix1___classification_criteria”.

Addition (Judgement)

Column G (appendix1___classification_criteria)

Reviewer

5. Please clarify what Cephem is? I am not familiar with this from North America. In the appendix listed as a nasal spray but in the manuscript it is treated as a systemic antibiotic. If a topical spray I suggest separating out from the systemic antimicrobials

Response

Thank you for bringing this matter to our attention. We have added the following information to the revised manuscript as suggested.

Changes (marked_manuscript)

→ Oral chem

Reviewer

6. Can the authors clarify what proportion of “inappropriate prescriptions” were for incorrect dose and which were unnecessary?

Response

Thank you for bringing this matter to our attention. There were no prescriptions with incorrect dosage and administration. However, a prescription was provided to a patient with a history of hypersensitivity to β2 stimulants (which is a contraindication)" (n=2). This information has been added to the Results section.

When medications are prescribed despite a lack of symptoms, we did not generally judge them as unnecessary (potentially inappropriate prescriptions), considering the possibility that the prescribing physician was anticipating future symptoms based on the patient's progress. We judged the appropriateness of the prescription drug based on the overall risk of side effects of the prescription drug and the patient's background.

Changes (marked_manuscript)

P.13, L225 (Results_ Details of Inappropriate Prescription)

There were two contraindicated doses (for patients with a history of β2-stimulant hypersensitivity), and the pharmacist posed questions regarding the prescription. There were no dosage errors．

Reviewer

7. Line 126: I am not familiar with “Kampo medicine”, can the authors briefly explain this exclusion?

Response

Thank you for bringing this matter to our attention. Chinese herbal medicines (Kampo medicines) are a combination of herbal medicines made from a number of plants and minerals. Originally developed in China, it was introduced in Japan and developed into a unique treatment method in Japan. Through thousands of years of experience, the effects of various combinations of herbal medicines have been confirmed. However, Kampo medicines contain thousands of compounds, and it is rarely clear which ingredients exert what effects. In addition, Kampo medicines were excluded from our study as little Western medical evidence is available.

Changes (Method_ Exclusion Criteria)

P6, L124 (marked_manuscript)

Since there is not enough scientific evidence for Chinese herbal medicines, also called “Kampo medicines”, prescriptions containing Kampo medicines were excluded from this study. Kampo medicine is a combination of herbal medicines made from a number of plants and minerals. Originally developed in China, it was introduced in Japan and developed into a unique treatment method in Japan. Through thousands of years of experience, the effects of various combinations of herbal medicines have been confirmed.

Reviewer

8. Line 142: I am not sure why the primary outcome was only presented descriptively and the secondary outcome was analyzed with a statistical test. With 136 participants it is a missed opportunity to not evaluate for predictors of inappropriate prescribing from the data collected. I recommend the authors perform bivariate and multivariable analysis evaluating predictors for inappropriate prescriptions.

Response

We thank the reviewers for their suggestions. The authors apologize for the lack of explanation, and at the same time, the authors agree with the opinion. Our current study was conducted as a challenging pilot study. Our most significant limitation was that we could not collect information on prescribing doctors and clinics from the IRB approval stage of the study, making causal inferences and comparative studies difficult. Therefore, primary outcomes were documented as descriptive instead of statistically analyzing a small sample size. In addition, the statistical analysis of the comparison of monetary values, which was different, has been deleted.

We appreciate your pertinent and constructive feedback.

Deletions (marked_manuscript)

P7, (Method_ Statistical Analyses)

Wilcoxon's rank-sum test was used to compare the median values of the quantitative data between the two groups. All statistically significant differences were defined at p-value <0.05.

P14, (Results_ Rates of Inappropriate Prescription and Medication Costs) 

showing a statistically significant difference in the median values of the two groups (p < 0.001).

Reviewer

9. I was surprised the authors did not capture whether a throat swab (rapid antigen or culture) was done for GAS. Is this common practice in primary care in Japan? Given how common sore throat as a complaint was I am not clear how antibiotic prescriptions were adjudicated without this information. Were all antibiotics assumed to be inappropriate? This relates back to point 4.

Response

Thank you very much for your valuable remarks. In primary care settings in Japan, the GAS pharyngeal swab is often omitted; the patient's risk of streptococcal infection is assessed based on the Center criteria, and if the risk is high, amoxicillin is generally prescribed without the GAS pharyngeal swab. In this case, when the pharmacist interviewed the patient, it was confirmed that the patient had been told by the physician that there was a high possibility of pharyngitis or streptococcal infection (appendix 2 item (13)) and if the patient has symptoms such as fever and sore throat and amoxicillin is prescribed, it is considered an appropriate prescription.

Reviewer

10. Line 200: Are these percentages of antibacterial agents the percent of all prescriptions? Percent of antimicrobials? Or percent of patients?

Response

Thank you for your questions. The relevant section describes the percentage of all prescriptions. We have changed the description in the revised manuscript as follows:

Changes (Results_ Details of Prescriptions)

P10, L199 (marked_manuscript)

Regarding antimicrobial agents, 44.9% of all prescriptions contained oral cephem (cefcapene pivoxil, cefdinir), 25.0% contained new quinolones (galenoxacin, levofloxacin), 9.5% contained macrolides (azithromycin, clarithromycin), and 2.2% contained Penicillin (amoxicillin) (Table 3).

Reviewer

11. As above comment if Cephem is not a systemic antibiotic consider separating this out from the other antibiotics

Response

Thank you for your valuable suggestion. We have separated the descriptions regarding Cephem to ensure that oral and topical sprays can be distinguished.

Changes (marked_manuscript) 

→ oral cephem

Reviewer

12. Table 3 lists the percent of prescriptions. Can you add the percent inappropriate to this table? For example 41.9% of prescriptions were H1 blockers (or is it 41.9% of patients received H1 blocker?)…what percent of those 57 H1 blockers were inappropriate?

Response

Thank you for bringing this matter to our attention. The percent inappropriateness has been added to Table 3 (Breakdown of potentially inappropriate prescription drugs for each drug prescribed).

Changes (marked_manuscript)

 P11-12 (Result_ Table 3. Details of prescription drugs (n = 136))

Reviewer

13.Can you convert Yen to US dollars in brackets?

Response

Thank you for bringing this matter to our attention. We have added the following information to the revised manuscript as suggested.

Changes (marked_manuscript)

P1, L20-21: 602.0 yen (IQR, 479.7–839.2)［$5.2 (IQR, 4.2–7.3) and 406.7 yen (IQR, 194.5–537.2) [$3.5 (IQR,1.7–4.7)], 

P13, L232-: The median total cost of the prescribed drugs was 593.6 yen (IQR, 470–795.6)［$5.2 (IQR, 4.1–6.9). In particular, the median cost of drugs in the inappropriate prescription group was 602.0 yen (IQR, 479.7–839.2)［$5.2 (IQR, 4.2–7.3), and the median cost of drugs in the appropriate prescription group was 406.7 yen (IQR, 194.5–537.2) [$3.5 (IQR,1.7–4.7)]

P17, L320-352: The median drug costs of inappropriate prescriptions and appropriate were 602.0 yen (IQR, 479.7–839.2)［$5.2 (IQR, 4.2–7.3) and 406.7 yen (IQR, 194.5–537.2) [$3.5 (IQR,1.7–4.7)] 

Reviewer

14. Table 4: Suggest combine with table 3 and include numbers (not just percentages). How many prescriptions, what percent of total, and what number and percent were inappropriate

Response

Thank you for your valuable suggestion. The number of prescriptions for each drug, overall percentage, number of inappropriate prescriptions, and percentage has been added to Table 3.

Changes (marked_manuscript)

 P11-12 (Result_ Table 3. Details of prescription drugs (n = 136))

Reviewer

15. As per point 8 suggest include a statistical evaluation of predictors of inappropriateness

Response

Thank you for your pertinent comments. As you mentioned, it is crucial to conduct multivariate analysis to evaluate predictors of inappropriate prescribing. However, due to the socio-ethical background of this study, we were not able to incorporate the essential physician and clinic characteristics as factors from the beginning. To the best of our knowledge, no previous survey has been conducted on the actual status of prescription medications, including symptomatic medications, for cold symptoms. Therefore, we conducted a pilot study on the actual status of prescription drugs in a limited area and with limited sample size. Next, based on the results of this study, we are considering conducting another survey on a national scale to evaluate predictors and conduct a multivariate analysis.

Reviewer

16. Line 260: clarify if Cephem is topical or systemic

Response

Thank you for your valuable suggestion. We have clarified whether the Cepham used was oral or nasal spray in the manuscript.

Changes (marked_manuscript)

P12, Table 3: Cephem (Oral) 

P12, Table 3: Cephem (Nasal spray)

Reviewer

17. Line 261: “Penicillin is recommended as the first choice…” how was this factored into the study? It is not clear how penicillin was adjudicated in this study (appropriate or inappropriate?)

Response

Thank you for bringing this matter to our attention. We confirmed during the interview that the patients had symptoms such as fever and sore throat, and the doctor explained that the patient had pharyngitis or streptococcal infection (appendix 2 item (13)). If amoxicillin was prescribed at that time, it was determined as an appropriate prescription.

Reviewer

18. Line 273: The provided reason for classifying quinolones as inappropriate is not accurate. Quinolones are inappropriate because they have no effect on viruses which cause the common cold. You have listed 1 rare but important side effect of that medication class.

Response

Thank you for your valuable suggestion. We have stated that quinolone antibacterials are inappropriate as they are ineffective against viruses that cause common colds.

Changes (Discussion_ The Rationale for Classification as the Potentially Inappropriate Prescribing Group)

P15, L266 (marked_manuscript)

New quinolones are considered inappropriate as they are ineffective against viruses which cause the common cold.

Reviewer

19. Line 280: I don’t understand the separation of sections titled “Factors leading…” and “Factors responsible…” The authors discuss one potential aspect leading to inappropriate prescribing (medical education). I agree this may be a factor but there is a whole behavioral science literature of the many complex reasons for inappropriate prescribing – perceived patient expectations, fear, habit, etc. Many reasons beyond education and knowledge base. While this may be beyond the scope of this manuscript to discuss in detail these are important to briefly discuss as the solutions to this complex problem will involve much more than modifying medical education or continuing medical education as we know these interventions, while important, have overall limited impact on behavior change and quality improvement.

Response

Thank you for bringing this matter to our attention. “Factors leading... " and "Factors responsible... " have been merged in the revised manuscript. We have also added the following sentence to the revised manuscript:

Changes (Discussion_ Factors Responsible for Prescriptions of Potentially Inappropriate Drugs)

P16, L301 (marked_manuscript)

However, it is unlikely that any single solution will alter the prescribing habits of physicians [42]. Cultural factors may also be involved in addressing this problem, and steps other than continuing medical and patient education are necessary.

Reviewer

20. Line 291: The authors summarize literature on drivers of inappropriate drugs but missed an opportunity in this study to contribute to that literature. As per point 8 and 15 above I suggest you add that evaluation.

Response

Thank you for your valuable comment. As you mentioned, it is crucial to conduct multivariate analysis to evaluate predictors of inappropriate prescribing. However, due to the socio-ethical background of this study, we were not able to incorporate the essential physician and clinic characteristics as factors from the beginning. To the best of our knowledge, no previous survey has been conducted on the actual status of prescription medications, including symptomatic medications, for cold symptoms. Therefore, we conducted a pilot study on the actual status of prescription drugs in a limited area and with limited sample size. Next, based on the results of this study, it is extremely important to conduct another survey on a nationwide scale, evaluate predictors, and conduct multivariate analysis.

Reviewer

21. Line 321: Suggest add to limitations that the findings are limited to adults without underlying medical comorbidities (as per study inclusion criteria)

Response

Thank you for your suggestion. The following information has been added to the Limitation section:

Addition (Discussion_ Limitaion)

P19, L355 (marked_manuscript)

Lastly, this study was limited to adult patients with no underlying medical conditions. The content of prescriptions for common cold patients is likely to be affected by the presence or absence of underlying diseases and the age of the patients. Therefore, the results of this study may not fully reflect the current status of prescribing for common cold symptoms in Japan.

We would like to express our sincere gratitude for your constructive feedback on our paper. We sincerely hope that the revised manuscript is suitable for publication.

---

## [Decision Letter · Decision Letter 1]

16 Feb 2022

PONE-D-21-33168R1Survey potentially inappropriate prescriptions for common cold symptoms in Japan: A prospective observational studyPLOS ONE

Dear Dr. Watari,

Thank you for submitting your manuscript to PLOS ONE. After careful consideration, we feel that it has merit but does not fully meet PLOS ONE’s publication criteria as it currently stands. Therefore, we invite you to submit a revised version of the manuscript that addresses the points raised during the review process.

Minor comments are still necessary in the present form.See the Reviewer's suggestions carefully and respond them appropriately.

We look forward to receiving your revised manuscript.

Kind regards,

Masaki Mogi

Academic Editor

PLOS ONE

Journal Requirements:

Reviewers' comments:

Reviewer's Responses to Questions

**Comments to the Author**

1. If the authors have adequately addressed your comments raised in a previous round of review and you feel that this manuscript is now acceptable for publication, you may indicate that here to bypass the “Comments to the Author” section, enter your conflict of interest statement in the “Confidential to Editor” section, and submit your "Accept" recommendation.

Reviewer #1: (No Response)

Reviewer #2: All comments have been addressed

2. Is the manuscript technically sound, and do the data support the conclusions?

Reviewer #1: Partly

Reviewer #2: Yes

3. Has the statistical analysis been performed appropriately and rigorously? 

Reviewer #1: N/A

Reviewer #2: Yes

4. Have the authors made all data underlying the findings in their manuscript fully available?

Reviewer #1: Yes

Reviewer #2: No

5. Is the manuscript presented in an intelligible fashion and written in standard English?

Reviewer #1: Yes

Reviewer #2: Yes

6. Review Comments to the Author

Reviewer #1: Reviewer

1. Method

I would suggest adding more in the Statistical analyses section on the statistical methods used for deriving results described in the results section (including ones included in the Tables). For example, currently there is no mention on IQRs in the statistical analyses section – please add an explanation on this.

Response:

Changes (Method_Statistical Analyses)

P7, L148

Dear author

Your comment has been addressed.

Reviewer

2. Results

How many physicians have prescribed the drugs in this time? What are the specialties of the doctors who prescribed it? What percentage of prescribing physicians belong to a hospital or clinic? Please add their characteristics if possible.

Response

Addition (Discussion_ Limitation)

P18, L347 to P19, L359 (marked_manuscript)

Dear author

Your comment has been addressed, but please make it more briefly and more compact.

Reviewer

3.Discussion

1) Four issues on the part of physicians have been pointed out as factors in inappropriate prescribing (Cephem antibacterials, β2-stimulants, and new quinolone antibacterials). What about the role of pharmacists, such as their intervention against inappropriate prescriptions? For example, in Japan, as a countermeasure against polypharmacy, a pharmaceutical management fee has been newly established as a dispensing fee, and pharmacists are encouraged to propose drug reduction.

Since there is no pharmacist's view for inappropriate prescribing in this document, please add their perspective.

Response:

Addition (Discussion_ Pharmacists' views on potentially inappropriate prescription)

P17, L306

Dear author

I understand it. However, I think the following point is one of the major problem faced by pharmacists.　This is because pharmacists in Japan have not yet established the professional competence to review and suggest prescriptions for pharmacotherapy.

Reviewer

3.Discussion

2) It has been shown that medication costs are higher in the inappropriate prescription

group, but this point was not good enough explained, though there are some words about Japanese healthcare system in the “Discussion”

Response

n_ Considerations for drug costs)

P17, L322 (marked_manuscript)

Dear author

I don't think the following explanation adequately describes the actual situation in Japan.　"Japan's universal health insurance system covers all prescription drugs without restrictions.”

For example, there is the Diagnosis Procedure Combination (DPC), and there is a system for polypharmacy, which provides reimbursement for the dispensing of drugs by reducing the number of drugs. Therefore, how about "Japan's universal health insurance system covers all prescription drugs generally.” ?

Reviewer

3.Discussion

3)It was described the first factor is that patients may be seeking medications from doctors in this document. However, some authors interviewed the patients actually. Did they confirm this factor? What was the patients' view?

From the perspective of “Choosing Wisely”, how do the authors think about the health literacy education for patients?

Response

Dear author

How about the following references as citations of Choosing Wisely?

Consumer Reports

https://www.consumerreports.org/doctors/questions-to-ask-your-doctor/

or

Muscat DM, et al. Evaluation of the Choosing Wisely Australia 5 Questions resource and a shared decision-making preparation video: protocol for an online experiment. BMJ Open 2019;9:e033126

https://bmjopen.bmj.com/content/9/11/e033126

Dear author

Finally, after reading the submitted paper more precisely again, I would like to mention an additional point.

The design of this study is a cross sectional study, and thus, it seems inappropriate to refer to this study as a prospective observational study. The reason is as follows; it is cross sectional because they looked at the prevalence of whether or not there had been an inappropriate prescription, where the survey lasted for 3 months – but in fact, the study participants were not followed for 3 months. It is recommended that the authors consult an epidemiologist.

Reviewer #2: The authors have done an excellent job responding to all reviewer comments. Everything has been adequately addressed. I have no further feedback. Thank you.

7. PLOS authors have the option to publish the peer review history of their article (what does this mean?). If published, this will include your full peer review and any attached files.

Reviewer #1: No

Reviewer #2: No

---

## [Author Response · Author response to Decision Letter 1]

4 Mar 2022

March 4, 2022

Dear PLOS ONE Editor:

Thank you for giving us the opportunity to revise our manuscript. The constructive suggestions and feedback provided by the Reviewer have enabled us to substantially improve the quality of our paper. We have provided a point-by-point response to the Reviewer’s suggestions. The revised portions have been indicated in green highlight in the revised manuscript.

Once again, thank you for your thorough and supportive peer review.

On behalf of all the authors, 

Yours sincerely,

Corresponding author

Takashi Watari, MD, MHQS, MCTM, Ph.D 

Shimane University Hospital, General Medicine Center, Shimane, Japan

89-1, Enya-cho, Izumo shi, Shimane, 693-8501, Japan

e-mail: wataritari@gmail.com; Phone: +81-853-20-2005; Fax: +81-853-20-2375

 

Reviewer 

2. Results

How many physicians have prescribed the drugs in this time? What are the specialties of the doctors who prescribed it? What percentage of prescribing physicians belong to a hospital or clinic? Please add their characteristics if possible.

Response

Addition (Discussion_ Limitation)

P18, L347 to P19, L359 (marked_manuscript)

Dear author

Your comment has been addressed, but please make it more briefly and more compact.

Response: 

We appreciate and agree with the reviewers’ comment. The following changes have been made in the revised manuscript:

Change (Discussion_ Limitation)

Page 19, Lines 362 to 367 (marked_manuscript)

Limitation: Sixth, the details regarding the prescribing physician were not investigated in this study as the Ethics Committee stipulates that "the content that identifies the prescribing physician (specialty, affiliation, and the number of individuals) must not be added to the content of the interview because it may cause disadvantages to the prescribing physician.” We believe that the prescribing physician's details are important factors and further research is needed.

Reviewer

Reviewer

3.Discussion

2) It has been shown that medication costs are higher in the inappropriate prescription

group, but this point was not good enough explained, though there are some words about Japanese healthcare system in the “Discussion”

Response

Discussion_ Considerations for drug costs)

P17, L322 (marked_manuscript)

Dear author

I don't think the following explanation adequately describes the actual situation in Japan.　"Japan's universal health insurance system covers all prescription drugs without restrictions.”

For example, there is the Diagnosis Procedure Combination (DPC), and there is a system for polypharmacy, which provides reimbursement for the dispensing of drugs by reducing the number of drugs. Therefore, how about "Japan's universal health insurance system covers all prescription drugs generally.” ?

Response

We appreciate and agree with the reviewers’ comment. The following changes have been made in the revised manuscript:

Addition (Discussion_ Considerations for drug costs)

Page 18, Line 331 (marked_manuscript)

Considerations for drug costs: Japan's universal health insurance system covers all prescription drugs generally.

Reviewer

Reviewer

3.Discussion

3)It was described the first factor is that patients may be seeking medications from doctors in this document. However, some authors interviewed the patients actually. Did they confirm this factor? What was the patients' view?

From the perspective of “Choosing Wisely”, how do the authors think about the health literacy education for patients?

Response

Dear author

How about the following references as citations of Choosing Wisely?

Consumer Reports

https://www.consumerreports.org/doctors/questions-to-ask-your-doctor/

or

Muscat DM, et al. Evaluation of the Choosing Wisely Australia 5 Questions resource and a shared decision-making preparation video: protocol for an online experiment. BMJ Open 2019;9:e033126

https://bmjopen.bmj.com/content/9/11/e033126

Response

We appreciate and agree with the reviewers’ comment. The following changes have been made in the revised manuscript:

Addition (Discussion_ Considerations for drug costs)

Page 18, Line 334 to 341 (marked_manuscript)

Considerations for drug costs: As part of the Choosing Wisely campaign, it is generally recommended that patients ask their doctors the following five questions [46]. (1) Do I really need this test or procedure? (2) What are the risks? (3) Are there simpler, safer options? (4) What happens if I do not do anything? (5) What are the costs? Patients must be aware of their own "treatment" by referring to the above questions to promote the appropriate use of medicines in Japan. It is necessary to incorporate education and measures that consider the interactive opinions of the medical professionals and patients.

Reviewer

Finally, after reading the submitted paper more precisely again, I would like to mention an additional point.

The design of this study is a cross sectional study, and thus, it seems inappropriate to refer to this study as a prospective observational study. The reason is as follows; it is cross sectional because they looked at the prevalence of whether or not there had been an inappropriate prescription, where the survey lasted for 3 months – but in fact, the study participants were not followed for 3 months. It is recommended that the authors consult an epidemiologist.

Response

We appreciate and agree with the reviewers’ comment. The following changes have been made in the revised manuscript:

Changes (marked_manuscript)

Page 1, Line 2: A Cross-Sectional Study

Page 1, Line 28: cross-sectional study

Page 4, Line 82: cross-sectional study

Page 14-15, Line 251-252: cross-sectional study

---

## [Decision Letter · Decision Letter 2]

10 Mar 2022

Survey of potentially inappropriate prescriptions for common cold symptoms in Japan: A cross-sectional study

PONE-D-21-33168R2

Dear Dr. Watari,

We’re pleased to inform you that your manuscript has been judged scientifically suitable for publication and will be formally accepted for publication once it meets all outstanding technical requirements.

Kind regards,

Masaki Mogi

Academic Editor

PLOS ONE

Additional Editor Comments (optional):

No further comment.

Reviewers' comments:

Reviewer's Responses to Questions

**Comments to the Author**

1. If the authors have adequately addressed your comments raised in a previous round of review and you feel that this manuscript is now acceptable for publication, you may indicate that here to bypass the “Comments to the Author” section, enter your conflict of interest statement in the “Confidential to Editor” section, and submit your "Accept" recommendation.

Reviewer #1: All comments have been addressed

2. Is the manuscript technically sound, and do the data support the conclusions?

Reviewer #1: Yes

3. Has the statistical analysis been performed appropriately and rigorously? 

Reviewer #1: Yes

4. Have the authors made all data underlying the findings in their manuscript fully available?

Reviewer #1: (No Response)

5. Is the manuscript presented in an intelligible fashion and written in standard English?

Reviewer #1: Yes

6. Review Comments to the Author

Reviewer #1: The changes made by the authors are well noted. It is appreciated that the comments and suggestions have been incorporated into the revised version.

7. PLOS authors have the option to publish the peer review history of their article (what does this mean?). If published, this will include your full peer review and any attached files.

Reviewer #1: No

---

## [Editor Report · Acceptance letter]

15 Mar 2022

PONE-D-21-33168R2 

Survey of potentially inappropriate prescriptions for common cold symptoms in Japan: A cross-sectional study 

Dear Dr. Watari:

I'm pleased to inform you that your manuscript has been deemed suitable for publication in PLOS ONE. Congratulations! Your manuscript is now with our production department. 

Kind regards, 

on behalf of

Dr. Masaki Mogi 

Academic Editor

PLOS ONE